# Gaussian Approximation and Multiplier Bootstrap for Stochastic Gradient Descent

## Abstract

In this paper, we establish the non-asymptotic validity of the multiplier bootstrap procedure for constructing the confidence sets using the Stochastic Gradient Descent (SGD) algorithm. Under appropriate regularity conditions, our approach avoids the need to approximate the limiting covariance of Polyak-Ruppert SGD iterates, which allows us to derive approximation rates in convex distance of order up to $1/\sqrt{n}$. Notably, this rate can be faster than the one that can be proven in the Polyak-Juditsky central limit theorem. To our knowledge, this provides the first fully non-asymptotic bound on the accuracy of bootstrap approximations in SGD algorithms. Our analysis builds on the Gaussian approximation results for nonlinear statistics of independent random variables.

## 1 Introduction

Stochastic Gradient Descent (SGD) is a widely used first-order optimization method that is well suited for large data sets and online learning. The algorithm has attracted significant attention; see [34, 31, 27, 26, 8]. SGD aims to solve the optimization problem:

$$f(\theta) \to \min_{\theta \in \mathbb{R}^d} , \qquad \nabla f(\theta) = \mathbb{E}_{\xi \sim \mathbb{P}_\xi}[F(\theta, \xi)] , \tag{1}$$

where $\xi$ is a random variable defined on a measurable space $(\mathsf{Z}, \mathcal{Z})$. Instead of the exact gradient $\nabla f(\theta)$, the algorithm accesses only unbiased stochastic estimates $F(\theta, \xi)$.

Throughout this work, we focus on the case of strongly convex objective functions and denote by $\theta^\star$ the unique minimizer of (1). The iterates $\theta_k$, $k \in \mathbb{N}$, generated by SGD follow the recursive update:

$$\theta_{k+1} = \theta_k - \alpha_{k+1} F(\theta_k, \xi_{k+1}) , \quad \theta_0 \in \mathbb{R}^d , \tag{2}$$

where $\{\alpha_k\}_{k \in \mathbb{N}}$ is a sequence of step sizes (or learning rates), which may be either diminishing or constant, and $\{\xi_k\}_{k \in \mathbb{N}}$ is an i.i.d. sequence sampled from $\mathbb{P}_\xi$. Theoretical properties of SGD, particularly in the convex and strongly convex settings, have been extensively studied; see, e.g., [28, 26, 8, 23]. Many optimization algorithms build upon the recurrence (2) to accelerate the convergence of the sequence $\theta_k$ to $\theta^\star$. Notable examples include momentum acceleration [32], variance reduction techniques [12, 39], and averaging methods. In this work, we focus on Polyak-Ruppert averaging, originally proposed in [36] and [31], which improves convergence by averaging the SGD iterates (2). Specifically, the estimator is defined as

$$\bar{\theta}_n = \frac{1}{n} \sum_{i=0}^{n-1} \theta_i , \quad n \in \mathbb{N} . \tag{3}$$

It has been established (see [31, Theorem 3]) that under appropriate conditions on the objective function $f$, the noisy gradient estimates $F$, and the step sizes $\alpha_k$, the sequence of averaged iterates

$\{\bar{\theta}_n\}_{n \in \mathbb{N}}$ is asymptotically normal:

$$\sqrt{n}(\bar{\theta}_n - \theta^\star) \overset{d}{\to} \mathcal{N}(0, \Sigma_\infty) , \tag{4}$$

where $\overset{d}{\to}$ denotes convergence in distribution, and $\mathcal{N}(0, \Sigma_\infty)$ is a zero-mean Gaussian distribution with covariance matrix $\Sigma_\infty$, defined later in Section 2.2. This result raises two key questions:

(i) what is the rate of convergence in (4)?

(ii) how can (4) be leveraged to construct confidence sets for $\theta^\star$, given that $\Sigma_\infty$ is unknown in practice?

In our paper we aim to answer both questions. To quantify convergence rates in (4), we employ convex distance, which is defined for random vectors $X, Y \in \mathbb{R}^d$ as

$$\mathsf{d}_\mathsf{C}(X, Y) = \sup_{B \in \mathsf{C}(\mathbb{R}^d)} \left| \mathbb{P}(X \in B) - \mathbb{P}(Y \in B) \right| ,$$

where $\mathsf{C}(\mathbb{R}^d)$ denotes the collection of convex subsets of $\mathbb{R}^d$. The authors of [42] derive Berry-Esseen-type bounds for $\mathsf{d}_\mathsf{C}(\sqrt{n}\Sigma_n^{-1/2}(\bar{\theta}_n - \theta^\star), \mathcal{N}(0, \mathrm{I}_d))$, where $\Sigma_n$ is the covariance matrix of the linearized counterpart of (2), see precise definitions (13) . We complement this result with the rates of convergence in (4). Interestingly, we also provide the lower bounds on the convex distance $\mathsf{d}_\mathsf{C}(\sqrt{n}(\bar{\theta}_n - \theta^\star), \mathcal{N}(0, \Sigma_\infty))$, which indicates, that for some choice of step sizes $\alpha_k$ in (2), the normal approximation by $\mathcal{N}(0, \Sigma_\infty)$ is less accurate, compared to normal approximation with other covariance matrix, in particular, with $\Sigma_n$. This effect has been previously observed in the bootstrap method for i.i.d. observations without the context of gradients methods, see [41, Theorem 3.11].

One of the popular approaches for solving (ii) is based on the *plug-in* methods [11, 9], which aim to construct an estimator $\hat{\Sigma}_n$ of $\Sigma_\infty$ directly. These methods often provide a non-asymptotic bounds on the closeness $\hat{\Sigma}_n$ is to $\Sigma_\infty$, often in terms of $\mathbb{E}[\|\hat{\Sigma}_n - \Sigma_\infty\|]$. At the same time, the analysis of this methods typically bypass the item (i) and the issues related with the rate of convergence in (4). In our paper we suggest, to the best of our knowledge, the first fully non-asymptotic analysis of procedure for constructing the confidence intervals, based on the bootstrap approach [15, 16], which avoids the direct approximation of $\Sigma_\infty$, moreover, theoretical analysis of the underlying procedure together with the results on normal approximation with $\mathcal{N}(0, \Sigma_\infty)$ from (i) shows that the same approximation rate can not be achieved by the plug-in methods, at least for some range of step sizes $\alpha_k$ in (2). Our key contributions are as follows:

- We establish the non-asymptotic validity of the multiplier bootstrap procedure introduced in [16]. Under appropriate regularity conditions, our bounds imply that the quantiles of the exact distribution of $\sqrt{n}(\bar{\theta}_n - \theta^\star)$ can be approximated, up to logarithmic factors, at a rate of $n^{-\gamma/2}$ for step sizes of the form $\alpha_k = c_0/(k + k_0)^\gamma, \gamma \in (1/2, 1)$. To our knowledge, this provides the first fully non-asymptotic bound on the accuracy of bootstrap approximations in SGD algorithms. Notably, this rate can be faster than the one that we can prove in (4). Our rates improve upon recent works [38, 46], which addressed the convergence rate in similar procedures for the LSA algorithm.

- Our analysis of the multiplier bootstrap procedure reveals an interesting property: unlike plug-in estimators, the validity of the bootstrap method does not directly depend on approximating $\sqrt{n}(\bar{\theta}_n - \theta^\star)$ by $\mathcal{N}(0, \Sigma_\infty)$. Instead, it requires approximating $\mathcal{N}(0, \Sigma_n)$ for some matrix $\Sigma_n$. The structure of $\Sigma_n$ and its associated convergence rates play a central role in our present analysis, both for convergence rate in (4) and non-asymptotic bootstrap validity. Precise definitions are provided in Section 2.2.

- We analyze the Polyak-Ruppert averaged SGD iterates (3) for strongly convex minimization problems and establish Gaussian approximation rates in (4) in terms of the convex distance. Specifically, we show that the approximation rate $\mathsf{d}_\mathsf{C}(\sqrt{n}(\bar{\theta}_n - \theta^\star), \mathcal{N}(0, \Sigma_\infty))$ is of order $n^{-1/4}$ when using the step size $\alpha_k = c_0/(k + k_0)^{3/4}$ with a suitably chosen $\alpha_0$. Our result is based on the techniques of [42] and [46]. We also provide the lower bound, which indicate that our rate of normal approximation with $\mathcal{N}(0, \Sigma_\infty)$ is tight in the regime $\alpha_k = c_0/(k + k_0)^\gamma$ with $\gamma \geq 3/4$.

**Notations.** Throughout this paper, we use the following notations. For a matrix $A \in \mathbb{R}^{d \times d}$ and a vector $x \in \mathbb{R}^d$, we denote by $\|A\|$ and $\|x\|$ their spectral norm and Euclidean norm, respectively. We also write $\|A\|_\mathrm{F}$ for the Frobenius norm of matrix $A$. Given a function $f : \mathbb{R}^d \to \mathbb{R}$, we write $\nabla f(\theta)$ and $\nabla^2 f(\theta)$ for its gradient and Hessian at a point $\theta$. Additionally, we use the standard abbreviations "i.i.d." for "independent and identically distributed" and "w.r.t." for "with respect to".

**Literature review.** Asymptotic properties of the SGD algorithm, including the asymptotic normality of the estimator $\bar{\theta}_n$ and its almost sure convergence, have been extensively studied for smooth and strongly convex minimization problems [31, 22, 7]. Optimal mean-squared error (MSE) bounds for $\theta_n - \theta^\star$ and $\bar{\theta}_n - \theta^\star$ were established in [27] for smooth and strongly convex objectives, and later refined in [26]. The case of constant-step size SGD for strongly convex problems has been analyzed in depth in [14]. High-probability bounds for SGD iterates were obtained in [33] and later extended in [20]. Both works address non-smooth and strongly convex minimization problems.

It is important to note that the results discussed above do not directly imply convergence rates for $\sqrt{n}(\bar{\theta}_n - \theta^\star)$ to $\mathcal{N}(0, \Sigma_\infty)$ in terms of $d_C(\cdot, \cdot)$ or the Kantorovich–Wasserstein distance. Among the relevant contributions in this direction, we highlight recent works [44, 38, 46], which provide quantitative bounds on the convergence rate in (4) for iterates of the temporal difference learning algorithm and general linear stochastic approximation (LSA) schemes. However, these algorithms do not necessarily correspond to SGD with a quadratic objective $f$, as the system matrix in LSA is not necessarily symmetric. Non-asymptotic convergence rates of order $1/\sqrt{n}$ in a smooth Wasserstein distance were established in [2]. Recent paper [1] provide Berry-Essen bounds for last iterate of SGD for high-dimensional linear regression of order up to $n^{-1/4}$.

Bootstrap methods for i.i.d. observations were first introduced in [15]. In the context of SGD methods, [16] proposed the multiplier bootstrap approach for constructing confidence intervals for $\theta^\star$ and established its asymptotic validity. The same algorithm, with non-asymptotic guarantees, was analyzed in [38] for the LSA algorithm, obtaining rate $n^{-1/4}$ when approximating quantiles of the exact distribution of $\sqrt{n}(\bar{\theta}_n - \theta^\star)$.

Popular group of methods for constructing confidence sets for $\theta^\star$ is based on estimating the asymptotic covariance matrix $\Sigma_\infty$. Plug-in and batch-mean estimators for $\Sigma_\infty$ attracted lot of attention, see [11, 9, 10], especially in the setting when the stochastic estimates of Hessian are available. The latter two papers focused on learning with contextual bandits. Estimates for $\Sigma_\infty$ based on batch-mean method and its online modification were considered in [11] and [49]. The authors in [25] considered the asymptotic validity of the plug-in estimator for $\Sigma_\infty$ in the local SGD setting. [47] refined the validity guarantees for both the multiplier bootstrap and batch-mean estimates of $\Sigma_\infty$ for nonconvex problems. However, these papers typically provide recovery rates $\Sigma_\infty$, but only show asymptotic validity of the proposed confidence intervals. A notable exception is the recent paper [46], where the temporal difference (TD) learning algorithm was studied. The authors of [46] provided purely non-asymptotic analysis of their procedure, obtaining the approximation rate $n^{-1/3}$ for quantiles of $\sqrt{n}(\bar{\theta}_n - \theta^\star)$.

## 2 Main results

This section establishes the nonasymptotic validity of the multiplier bootstrap method proposed in [16]. We focus on smooth and strongly convex minimization problems, following the framework established in [26], [2] and [42]. The underlying procedure is based on perturbing the trajectory (2). We restate the procedure for the sake of clarity. Let $\mathcal{W}^{n-1} = \{w_\ell\}_{1 \le \ell \le n-1}$ be i.i.d. random variables with distribution $\mathbb{P}_w$, each with mean $\mathbb{E}[w_1] = 1$ and variance $\mathrm{Var}[w_1] = 1$. Assume $\mathcal{W}^{n-1}$ is independent of $\Xi^{n-1} = \{\xi_\ell\}_{1 \le \ell \le n-1}$. We then use $\mathcal{W}^{n-1}$ to construct randomly perturbed SGD trajectories, following the same recursive structure as the primary sequence

$$
\begin{aligned}
\theta_k^{\mathsf{b}} &= \theta_{k-1}^{\mathsf{b}} - \alpha_k w_k \{\nabla f(\theta_{k-1}^{\mathsf{b}}) + g(\theta_{k-1}^{\mathsf{b}}, \xi_k) + \eta(\xi_k)\} , \quad k \ge 1 , \quad \theta_0^{\mathsf{b}} = \theta_0 , \\
\bar{\theta}_n^{\mathsf{b}} &= n^{-1} \sum_{k=0}^{n-1} \theta_k^{\mathsf{b}} , \quad n \ge 1 .
\end{aligned}
\tag{5}
$$

Note that, when generating different weights $w_k$, we can draw samples from the conditional distribution of $\bar{\theta}_n^{\mathsf{b}}$ given the data $\Xi^{n-1}$. We further denote $\mathbb{P}^{\mathsf{b}} = \mathbb{P}(\cdot \mid \Xi^{n-1})$ and $\mathbb{E}^{\mathsf{b}} = \mathbb{E}(\cdot \mid \Xi^{n-1})$.

The core principle behind the bootstrap procedure (5) is that the "bootstrap world" probabilities $\mathbb{P}^{\mathsf{b}}\big(\sqrt{n}(\bar{\theta}_n^{\mathsf{b}} - \bar{\theta}_n) \in B\big)$ are close to $\mathbb{P}\big(\sqrt{n}(\bar{\theta}_n - \theta^\star) \in B\big)$ for $B \in C(\mathbb{R}^d)$. More formally, we say that the procedure (5) is asymptotically valid if

$$
\sup_{B \in C(\mathbb{R}^d)} \left| \mathbb{P}^{\mathsf{b}}\big(\sqrt{n}(\bar{\theta}_n^{\mathsf{b}} - \bar{\theta}_n) \in B\big) - \mathbb{P}\big(\sqrt{n}(\bar{\theta}_n - \theta^\star) \in B\big) \right|
\tag{6}
$$

converges to 0 in $\mathbb{P}$-probability as $n \to \infty$. This result was studied in [16] under assumptions close to the original paper [31]. While an analytical expression for $\mathbb{P}^{\mathsf{b}}\big(\sqrt{n}(\bar{\theta}_n^{\mathsf{b}} - \bar{\theta}_n) \in B\big)$ is unavailable, it

can be approximated via Monte Carlo simulations by generating $M$ perturbed trajectories according to (5). Standard arguments (see, e.g., [40, Section 5.1]) suggest that the accuracy of this Monte Carlo approximation scales as $\mathcal{O}(M^{-1/2})$ when generating $M$ parallel perturbed trajectories in (5).

**Assumptions.** We impose the following regularity conditions on the objective function $f$:

**A1.** *The function $f$ is two times continuously differentiable and $L_1$-smooth on $\mathbb{R}^d$, i.e., there is a constant $L_1 > 0$, such that for any $\theta, \theta' \in \mathbb{R}^d$,*

$$\|\nabla f(\theta) - \nabla f(\theta')\| \le L_1 \|\theta - \theta'\| .$$

*Moreover, we assume that $f$ is $\mu$-strongly convex on $\mathbb{R}^d$, that is, there exists a constant $\mu > 0$, such that for any $\theta, \theta' \in \mathbb{R}^d$, it holds that*

$$(\mu/2)\|\theta - \theta'\|^2 \le f(\theta) - f(\theta') - \langle \nabla f(\theta'), \theta - \theta' \rangle .$$

A1 implies the following two-sided bound on the Hessian $\nabla^2 f(\theta)$, $\mu \mathrm{I}_d \preceq \nabla^2 f(\theta) \preceq L_1 \mathrm{I}_d$ for all $\theta \in \mathbb{R}^d$. We now formalize the assumptions on $F(\theta, \xi)$. Namely, we rewrite $F(\theta, \xi)$ as

$$F(\theta_{k-1}, \xi_k) = \nabla f(\theta_{k-1}) + \zeta_k,$$

where $\{\zeta_k\}_{k \in \mathbb{N}}$ is a sequence of $d$-dimensional random vectors. Then the SGD recursion takes form

$$\theta_k = \theta_{k-1} - \alpha_k (\nabla f(\theta_{k-1}) + \zeta_k) , \quad \theta_0 \in \mathbb{R}^d . \tag{7}$$

We impose the following assumption on the noise sequence $\zeta_k$:

**A2.** *For each $k \ge 1$, $\zeta_k$ admits the decomposition $\zeta_k = \eta(\xi_k) + g(\theta_{k-1}, \xi_k)$, where*

*(i)* $\{\xi_k\}_{k=1}^{n-1}$ *is a sequence of i.i.d. random variables on $(\mathsf{Z}, \mathcal{Z})$ with distribution $\mathbb{P}_\xi$, $\eta : \mathsf{Z} \to \mathbb{R}^d$ is a function such that $\mathbb{E}[\eta(\xi_1)] = 0$ and $\mathbb{E}[\eta(\xi_1)\eta(\xi_1)^\top] = \Sigma_\xi$. Moreover, $\lambda_{\min}(\Sigma_\xi) > 0$.*

*(ii) The function $g : \mathbb{R}^d \times \mathsf{Z} \to \mathbb{R}^d$ satisfies $\mathbb{E}[g(\theta, \xi_1)] = 0$ for any $\theta \in \mathbb{R}^d$. Moreover, there exists $L_2 > 0$ such that for any $\theta, \theta' \in \mathbb{R}^d$, it holds that*

$$\|g(\theta, \xi) - g(\theta', \xi)\| \le L_2 \|\theta - \theta'\| \quad and \quad g(\theta^\star, z) = 0 \text{ for all } z \in \mathsf{Z} . \tag{8}$$

*(iii) There exist $C_{1,\xi}, C_{2,\xi} > 0$ such that $\mathbb{P}_\xi$-almost surely that $\|\eta(\xi)\| \le C_{1,\xi}$ and $\sup_\theta \|g(\theta, \xi)\| \le C_{2,\xi}$.*

As an example of a sequence $\zeta_k$ satisfying conditions (i) and (ii) from A2, consider the case when the oracle function $F(\theta, \xi)$ satisfies:

1. $\mathbb{E}[F(\theta, \xi)] = \nabla f(\theta)$ for all $\theta \in \mathbb{R}^d$;
2. $\|F(\theta, \xi) - F(\theta', \xi)\| \le L\|\theta - \theta'\|$ for all $\xi \in \mathsf{Z}$, and $\sup_\theta |F(\theta, \xi) - F(\theta^\star, \xi)| \le c_\xi$ for some $c_\xi > 0$.

In this case, (i) and (ii) from A2 holds with $\eta(\xi) = F(\theta^\star, \xi)$ and $g(\theta, \xi) = F(\theta, \xi) - F(\theta^\star, \xi)$. Additionally, note that the identity (8) can be relaxed when one considers only last iterate bounds, such as $\mathbb{E}[\|\theta_k - \theta^\star\|^2]$, see [26]. Item (ii) from A2 is often imposed when considering averaged iterates, see [26, Assumption H2'], and [14, 42].

The assumption (iii) from A2 is crucial to prove high-order moment bounds (20), see Lemma 15. In our proof, we closely follow the argument presented in [20, Theorem 4.1], which requires that the noise variables $\zeta_k$ be almost sure to be bounded. This setting can be generalized to the case where $\zeta_k$ is sub-Gaussian conditioned on $\mathcal{F}_{k-1}$ with variance proxy which is uniformly bounded by a constant factor, that is, there is a constant $M$, such that $\mathbb{E}[\exp\{\|F(\theta, \xi_1)\|^2/M^2\}] \le 2$ for any $\theta \in \mathbb{R}^d$. This assumption is widely considered in the literature; see [27, 21], and the remarks in [20]. However, when $\zeta_k = g(\theta_{k-1}, \xi_k) + \eta(\xi_k)$ and $g$ is only Lipschitz w.r.t. $\theta$, its moments will naturally scale with $\|\theta_{k-1} - \theta^\star\|$, thus the sub-Gaussian bound with $M$ not depending upon $\theta$ is unlikely to hold. Other authors who considered bounds of type (20), e.g. [33], made stronger assumption that $\sup_{\theta \in \mathbb{R}^d} \|F(\theta, \xi)\|$ is a.s. bounded. Another popular direction is to consider schemes for gradient clipping; see e.g. [37]. Unfortunately, employing such schemes change the key representation (12) that we use later in the proof of the main result (see Theorem 1). We leave further studies of clipped gradient schemes for future work. We further impose condition on the Hessian matrix $\nabla^2 f(\theta)$ at $\theta^\star$:

**A3.** *There exist $L_3, \beta > 0$ such that for all $\theta$ with $\|\theta - \theta^\star\| \le \beta$, it holds*

$$\|\nabla^2 f(\theta) - \nabla^2 f(\theta^\star)\| \le L_3 \|\theta - \theta^\star\| .$$

A3 ensures that the Hessian of $f$ is Lipschitz continuous in a neighborhood of $\theta^*$. Similar assumptions have been previously considered in [42] and [2], as well as in other works on first-order optimization methods, see, e.g. [24]. Several studies on the non-asymptotic analysis of SGD impose stronger smoothness assumptions, such as bounded derivatives of $f$ up to order four, see [14]. We additionally assume an almost sure co-coercivity of the stochastic gradient:

**A4.** *The stochastic gradient $F(\theta, \xi) := \nabla f(\theta) + g(\theta, \xi) + \eta(\xi)$ is almost surely $L_4$-co-coercive, that is, for any $\theta, \theta' \in \mathbb{R}^d$, it holds $\mathbb{P}_\xi$-almost surely that*

$$L_4 \langle F(\theta, \xi) - F(\theta', \xi), \theta - \theta' \rangle \geq \|F(\theta, \xi) - F(\theta', \xi)\|^2 .$$

In particular, A4 holds (see e.g. [48]), when there is a function $v(\theta, \xi)$, such that $F(\theta, \xi) = \nabla_\theta v(\theta, \xi)$, where $v(\theta, \xi)$ is convex $\mathbb{P}_\xi$-a.s. and $L_4$-smooth. Co-coercivity is stronger than just requiring $F(\theta, \xi)$ to be monotone. We also impose an assumption on the bootstrap weights $W_i$ used in the algorithm:

**A5.** *There exist constants $0 < W_{\min} < W_{\max} < +\infty$, such that $W_{\min} \leq w_1 \leq W_{\max}$ a.s.*

The original paper [16] also considered positive bootstrap weights $w_i$. We have to impose bounded-ness of $w_i$ due to our high-probability bound on Lemma 15. A particular example of a distribution satisfying A5 is provided in Appendix E.1. We also consider the following bound for step sizes $\alpha_k$ and sample size $n$:

**A6.** *Let $\alpha_k = c_0 \{k_0 + k\}^{-\gamma}$, where $\gamma \in (1/2, 1)$, an $c_0$ satisfies $c_0 W_{\max} \max(2L_4, \mu) \leq 1$ and $k_0 \geq (\frac{2\gamma}{\mu c_0 W_{\min}})^{1/(1-\gamma)}$.*

**A7.** *Number of observations $n$ satisfies $n \geq \mathrm{e}^3$ and $\frac{n}{\log(2dn)} \geq \max(1, \frac{(20C_{Q,\xi}C_\Sigma^2)^2}{9})$, where the constants $C_{Q,\xi}$ and $C_\Sigma$ are defined in (60) and (27), respectively.*

The particular bound on $k_0$ in A6 appears due to the high-order moment bounds (see Lemma 15 in appendix). We note that it is possible to remove the co-coercivity assumption A4, but at the price of slightly stronger constraints on $c_0$ above. We discuss the bound on the number of observations imposed in A7 later in the proof of Theorem 1.

## 2.1 Non-asymptotic multiplier bootstrap validity

**Theorem 1.** *Assume A1 - A7. Then with $\mathbb{P}$ - probability at least $1 - 2/n$, it holds*

$$\sup_{B \in \mathsf{C}(\mathbb{R}^d)} |\mathbb{P}^{\mathsf{b}}(\sqrt{n}(\bar{\theta}_n^{\mathsf{b}} - \bar{\theta}_n) \in B) - \mathbb{P}(\sqrt{n}(\bar{\theta}_n - \theta^\star) \in B)| \leq \frac{\mathrm{C}_1 \sqrt{\log n}}{n^{1/2}} + \frac{\mathrm{C}_2 \log n}{n^{\gamma - 1/2}} + \frac{\mathrm{C}_3 (\log n)^{3/2}}{n^{\gamma/2}} ,$$

*where $\mathrm{C}_1, \mathrm{C}_2$ and $\mathrm{C}_3$ are given in Appendix E.7, equation (63).*

**Remark 1.** *It is possible to prove the result of Theorem 1 for the step size $\alpha_k = c_0/(k + k_0)$. The required Gaussian approximation result with the covariance matrix $\Sigma_n$ is proved in [42], and we expect that the only difference with Theorem 1 will occur in extra $\log n$ factors in the corresponding bound and slightly different conditions on $c_0$ and $k_0$ in A6.*

*Proof sketch of Theorem 1.* The proof of non-asymptotic bootstrap validity is based on the Gaussian approximation performed both in the "real" world and bootstrap world together with an appropriate Gaussian comparison inequality:

Real world: $\qquad \sqrt{n}(\bar{\theta}_n - \theta^\star) \xleftarrow{\text{Gaussian approximation, Th. 2}} \Sigma^{1/2} Y \sim \mathcal{N}(0, \Sigma)$

$\qquad\qquad\qquad\qquad\qquad\qquad\qquad\qquad\qquad\qquad\qquad\qquad\qquad\quad \Big\updownarrow \text{Gaussian comparison, Lem. 19}$

Bootstrap world: $\sqrt{n}(\bar{\theta}_n^{\mathsf{b}} - \bar{\theta}_n) \xleftarrow{\text{Gaussian approximation, Th. 3}} \{\Sigma^{\mathsf{b}}\}^{1/2} Y^{\mathsf{b}} \sim \mathcal{N}(0, \Sigma^{\mathsf{b}}) .$

Here $\Sigma$ and $\Sigma^{\mathsf{b}}$ are some covariance matrices to be chosen later. In order to understand where the Gaussian approximation comes from, we consider the process of linearization of statistics $\sqrt{n}(\bar{\theta}_n - \theta^\star)$ and $\sqrt{n}(\bar{\theta}_n^{\mathsf{b}} - \bar{\theta}_n)$. We provide details for $\sqrt{n}(\bar{\theta}_n - \theta^\star)$, and give similar derivations for $\sqrt{n}(\bar{\theta}_n^{\mathsf{b}} - \bar{\theta}_n)$ in Section 2.3. Denote $G = \nabla^2 f(\theta^\star)$. We expand $\sqrt{n}(\bar{\theta}_n - \theta^\star)$ into a weighted sum of independent random vectors, along with the remaining terms of smaller order. By the Newton-Leibniz formula, we obtain

$$\nabla f(\theta) = G(\theta - \theta^\star) + H(\theta), \tag{9}$$

where $H(\theta) = \int_0^1 (\nabla^2 f(\theta^\star + t(\theta - \theta^\star)) - G)(\theta - \theta^\star)\,\mathrm{d}t$. Note that $H(\theta)$ is of the order $\|\theta - \theta^\star\|^2$ (see Lemma 5). The recursion for the SGD algorithm error (7) can be expressed as

$$\theta_k - \theta^\star = (\mathrm{I}_d - \alpha_k G)(\theta_{k-1} - \theta^\star) - \alpha_k(\eta(\xi_k) + g(\theta_{k-1}, \xi_k) + H(\theta_{k-1}))\ . \qquad (10)$$

For $i \in \{0, \ldots, n-1\}$, we define the matrices

$$Q_i = \alpha_i \sum_{j=i}^{n-1} \prod_{k=i+1}^{j} (\mathrm{I}_d - \alpha_k G)\ , \qquad (11)$$

where empty products are defined to be equal to $\mathrm{I}_d$ by convention. Then taking average of (10) and rearranging the terms, we obtain the following expansion:

$$\sqrt{n}(\bar{\theta}_n - \theta^\star) = W + D\ , \quad W = -\frac{1}{\sqrt{n}} \sum_{i=1}^{n-1} Q_i \eta(\xi_i), \quad D = \sqrt{n}(\bar{\theta}_n - \theta^\star) - W\ . \qquad (12)$$

Note that $W$ is a weighted sum of i.i.d. random vectors with mean zero and covariance matrix

$$\Sigma_n = n^{-1} \sum_{k=1}^{n-1} Q_k \Sigma_\xi Q_k^\top\ , \qquad (13)$$

and $D$ is the remainder term which is defined in Appendix C, equation (39). Furthermore, in Appendix D.1 we show that $Q_i$ may be approximated by $G^{-\top}$ and $\Sigma_n$ approximates

$$\Sigma_\infty = G^{-1} \Sigma_\xi G^{-\top}\ .$$

We expect that the summand $D$ does not significantly distort the asymptotic distribution for the linear statistic $W$, which should be Gaussian by virtue of the central limit theorem. An important question is the choice of the approximating Gaussian distribution $\mathcal{N}(0, \Sigma)$ with $\Sigma = \Sigma_n$ or $\Sigma_\infty$ as well their bootstrap counterpart $\Sigma^{\mathrm{b}}$. This choice is instrumental in the sense that it does not change the procedure (5), but only affects the rates in (6). The authors of [16] choose the approximation with $\mathcal{N}(0, \Sigma_\infty)$ for their asymptotic analysis. A similar approach was considered in [38, Theorem 3] for the LSA algorithm setting. However, as it will be shown later in Theorem 4, this choice implies that the rate of normal approximation in (6) is not faster than $n^{-1/4}$. At the same time, Theorem 2 and Theorem 3 below demonstrate that we can achieve approximation rates of up to $n^{-1/2}$ by selecting $\Sigma = \Sigma_n$ in the diagram 2.1, and its bootstrap-world counterpart in the Gaussian approximation. To finish the proof, it remains to apply the Gaussian comparison inequality; see Lemma 19. Detailed proof of Theorem 1 is provided in Appendix E. □

**Discussion.** In [38] a counterpart of Theorem 1 was established with an approximation rate of the order $n^{-1/4}$ up to logarithmic factors for the setting of the LSA algorithm. The obtained rate is suboptimal, since the authors have chosen $\mathcal{N}(0, \Sigma_\infty)$ for Gaussian approximation when showing bootstrap validity. A recent paper [46] improved this rate to $n^{-1/3}$ for the temporal learning (TD) procedure with linear function approximation. The algorithm they considered is based on the direct estimate of $\Sigma_\infty$, yielding a rate of order $n^{-1/3}$ when approximating the quantiles of $\sqrt{n}(\bar{\theta}_n - \theta^\star)$, see [46, Theorem 3.4 and 3.5]. The authors in [11] constructed a plug-in estimator $\hat{\Sigma}_n$ of $\Sigma_\infty$ and showed guarantees of the form $\mathbb{E}[\|\hat{\Sigma}_n - \Sigma_\infty\|] \lesssim C n^{-\gamma/2}$, $\gamma \in (1/2, 1)$ under weaker assumptions than those considered in the current section. At the same time, approximating quantiles of $\sqrt{n}(\bar{\theta}_n - \theta^\star)$ with the method of [11] would require one more step - a Berry-Esseen type bound on the rate of approximation of $\sqrt{n}(\bar{\theta}_n - \theta^\star)$ with $\mathcal{N}(0, \Sigma_\infty)$. As we show in Theorem 4, this rate vanishes as $\gamma \to 1$, which introduces an additional trade-off to the potential analysis of the plug-in procedures based on estimating $\Sigma_\infty$. This effect highlights the fundamental difference between the multiplier bootstrap approach and the plug-in approach of [11].

Moreover, we highlight that in-expectation bound $\mathbb{E}[\|\hat{\Sigma}_n - \Sigma_\infty\|]$, which are typically studied in literature for plug-in estimates [11, 35], are not sufficient to prove the analogue of the Gaussian comparison result Lemma 1 for $\mathcal{N}(0, \hat{\Sigma}_n)$ and $\mathcal{N}(0, \Sigma_\infty)$ on the set with large $\mathbb{P}$-probability. Thus, the complete non-asymptotic analysis of the confidence sets constructed with the plug-in procedure, remains an open problem.

## 2.2 Gaussian approximation in the real world

For results of this section, assumptions A2 and A6 can be relaxed. We impose a family of assumptions, denoted as A8(p) with $p \geq 2$, on the noise sequence $\zeta_k$, and A9 on the step sizes $\alpha_k$:

**A 8** (p). *Conditions (i) and (ii) from A 2 holds. Moreover, there exists $\sigma_p > 0$ such that $\mathbb{E}^{1/p}[\|\eta(\xi_1)\|^p] \leq \sigma_p$ .*

**A9.** *Suppose that $\alpha_k = c_0/(k_0 + k)^\gamma$, where $\gamma \in (1/2, 1)$, $k_0 \geq 1$, and $c_0$ satisfies $2c_0 L_1 \leq 1$.*

Note that $A6$ implies A9, as well as A2 implies A8($p$) for any $p \geq 2$. In the main result of this section we provide the Gaussian approximation result for $\sqrt{n}(\bar{\theta}_n - \theta^\star)$ with $\mathcal{N}(0, \Sigma_n)$, which refines the bounds obtained in [42, Theorem 3.4] and is instrumental for further studies of normal approximation with $\mathcal{N}(0, \Sigma_\infty)$ in Section 2.4.

**Theorem 2.** *Assume A1, A3, A8(4), A9. Then, with $Y \sim \mathcal{N}(0, \mathrm{I}_d)$, it holds that*

$$d_{\mathsf{C}}(\sqrt{n}\Sigma_n^{-1/2}(\bar{\theta}_n - \theta^\star), Y) \leq \frac{\mathrm{C}_4}{\sqrt{n}} + \frac{\mathrm{C}_5}{n^{\gamma - 1/2}} + \frac{\mathrm{C}_6}{n^{\gamma/2}} \,, \tag{14}$$

*where $\mathrm{C}_4, \mathrm{C}_5, \mathrm{C}_6$ are given in Appendix C, equation (40). Moreover, since $\Sigma_n$ is non-degenerate, and an image of a convex set under non-degenerate linear mapping is a convex set, we have*

$$d_{\mathsf{C}}(\sqrt{n}\Sigma_n^{-1/2}(\bar{\theta}_n - \theta^\star), Y) = d_{\mathsf{C}}(\sqrt{n}(\bar{\theta}_n - \theta^\star), \Sigma_n^{1/2}Y) \,.$$

**Remark 2.** *When $\gamma \to 1$, the correction terms above scale as $\mathcal{O}(1/\sqrt{n})$, yielding the overall approximation rate that approaches $1/\sqrt{n}$. Expressions for $\mathrm{C}_4, \mathrm{C}_5, \mathrm{C}_6$ from Theorem 2 depend upon the problem dimension d, parameters specified in A1 - A8(4)- A3-A9. Moreover, $\mathrm{C}_5$ depends upon $\|\theta_0 - \theta^\star\|$. When $\gamma \in (0, 1)$, we have that $1/n^{\gamma/2} < 1/n^{\gamma - 1/2}$, thus, the term $\mathrm{C}_5 /n^{\gamma - 1/2}$ dominates. We prefer to keep both terms in (14), since they are responsible for the moments of statistics $\frac{1}{\sqrt{n}} \sum_{i=1}^{n-1} Q_i H(\theta_{i-1})$ and $\frac{1}{\sqrt{n}} \sum_{i=1}^{n-1} Q_i g(\theta_{i-1}, \xi_i)$, respectively. The first of them has non-zero mean, since $H(\theta_{i-1})$ is quadratic in $\|\theta_i - \theta^\star\|^2$. When using constant step size SGD, one can correct this term using the Richardson-Romberg technique [14, 43], however, it is unclear if this type of ideas can be generalized for diminishing step size.*

*Proof sketch of Theorem 2.* The decomposition (12) represents a particular instance of the general problem of Gaussian approximation for nonlinear statistics of the form $\sqrt{n}(\bar{\theta}_n - \theta^\star)$, where the estimator is expressed as the sum of linear and nonlinear components. To establish the Gaussian approximation result, we adapt the arguments from [42], which can be stated as follows. Let $X_1, \ldots, X_n$ be independent random variables taking values in some space $\mathcal{X}$, and let $T = T(X_1, \ldots, X_n)$ be a general $d$-dimensional statistic that can be decomposed as

$$W := W(X_1, \ldots, X_n) = \sum_{\ell=1}^n Z_\ell, \quad D := D(X_1, \ldots, X_n) = T - W \,.$$

Here, we define $Z_\ell = r_\ell(X_\ell)$, where $r_\ell : \mathcal{X} \to \mathbb{R}^d$ is a Borel-measurable function. The term $D$ represents the nonlinear component and is treated as an error term, assumed to be "small" relative to $W$ in an appropriate sense. Suppose that $\mathbb{E}[Z_\ell] = 0$ and that the $Z_\ell$ is normalized in such a way that $\sum_{\ell=1}^n \mathbb{E}[Z_\ell Z_\ell^\top] = \mathrm{I}_d$ holds. Let $\Upsilon_n = \sum_{\ell=1}^n \mathbb{E}[\|Z_\ell\|^3]$. Then, for $Y \sim \mathcal{N}(0, \mathrm{I}_d)$, the following bound holds:

$$d_{\mathsf{C}}(T, Y) \leq 259 d^{1/2}\Upsilon_n + 2\mathbb{E}[\|W\|\|D\|] + 2\sum_{\ell=1}^n \mathbb{E}[\|Z_\ell\|\|D - D^{(\ell)}\|], \tag{15}$$

where $D^{(\ell)} = D(X_1, \ldots, X_{\ell-1}, X_\ell', X_{\ell+1}, \ldots, X_n)$ and $X_\ell'$ is an independent copy of $X_\ell$. This result follows from [42, Theorem 2.1]. Furthermore, this bound can be extended to the case where $\sum_{\ell=1}^n \mathbb{E}[Z_\ell Z_\ell^\top] = \Sigma \succ 0$, as detailed in [42, Corollary 2.3]. In order to apply (15), we let $X_i = \xi_i$, $Z_\ell = h(X_\ell)$, $\xi_i'$ be an i.i.d. copy of $\xi_i$. Then we need to upper bound $\mathbb{E}^{1/2}[\|D(\xi_1, \ldots, \xi_{n-1})\|^2]$ and $\mathbb{E}^{1/2}[\|D - D_i'\|^2]$, respectively. Detailed proof is given in Appendix C. $\qquad\square$

## 2.3 Gaussian approximation in the bootstrap world

In the main result of this section, we study the Gaussian approximation result for $\sqrt{n}(\bar{\theta}_n^{\mathsf{b}} - \bar{\theta}_n)$ with appropriate normal distribution with respect to $\mathbb{P}^{\mathsf{b}}$. Despite this result is similar in its nature with the one of Theorem 2, it requires to handle some significant challenges that arises when working in the "bootstrap world". Our first steps are the same as in (10) and (5):

$$\begin{aligned}
\theta_k^{\mathsf{b}} - \theta_k = {}& (I - \alpha_k G)(\theta_{k-1}^{\mathsf{b}} - \theta_{k-1}) \\
& - \alpha_k \big(H(\theta_{k-1}^{\mathsf{b}}) + g(\theta_{k-1}^{\mathsf{b}}, \xi_k) - H(\theta_{k-1}) - g(\theta_{k-1}, \xi_k)\big) \\
& - \alpha_k(w_k - 1)\big(G(\theta_{k-1}^{\mathsf{b}} - \theta^\star) + \eta(\xi_k) + g(\theta_{k-1}^{\mathsf{b}}, \xi_k) + H(\theta_{k-1}^{\mathsf{b}})\big) \,.
\end{aligned} \tag{16}$$

293 Taking an average of (16) and rearranging the terms, we obtain a counterpart of (12):

$$\sqrt{n}(\bar{\theta}_n^{\mathsf{b}} - \bar{\theta}_n) = W^{\mathsf{b}} + D^{\mathsf{b}} \, ,$$

$$W^{\mathsf{b}} = -\frac{1}{\sqrt{n}} \sum_{i=1}^{n-1} (w_i - 1) Q_i \eta(\xi_i) \, , \quad D^{\mathsf{b}} = \sqrt{n}(\bar{\theta}_n^{\mathsf{b}} - \bar{\theta}_n) - W^{\mathsf{b}} \, . \tag{17}$$

294 Here $W^{\mathsf{b}}$ is a weighted sum of i.i.d. random variables $\Xi^{n-1}$, such that $\mathbb{E}^{\mathsf{b}}[W^{\mathsf{b}}] = 0$ and

$$\mathbb{E}^{\mathsf{b}}[W^{\mathsf{b}}\{W^{\mathsf{b}}\}^\top] := \Sigma_n^{\mathsf{b}} = n^{-1} \sum_{i=1}^{n-1} Q_i \eta(\xi_i) \eta(\xi_i)^\top Q_i^\top \, ,$$

295 and $D^{\mathsf{b}}$ is a non-linear statistic of $\Xi^{n-1}$. The principal difficulty arises when considering the
296 conditional distribution of $\sqrt{n}(\bar{\theta}_n^{\mathsf{b}} - \bar{\theta}_n)$ given the data $\Xi^{n-1}$. In fact, the approach of [42] would
297 require to control the second moments of $D^{\mathsf{b}}$ and $D^{\mathsf{b}} - \{D^{\mathsf{b}}\}^{(i)}$ with respect to a bootstrap measure
298 $\mathbb{P}^{\mathsf{b}}$, on the high-probability event with respect to a measure $\mathbb{P}$. At the same time, we loose a martingale
299 structure of the summands in $D^{\mathsf{b}}$, unless we condition on the extended filtration

$$\widetilde{\mathcal{F}}_i = \sigma(w_1, \dots w_i, \xi_1, \dots \xi_i) \, , 1 \leq i \leq n - 1 \, . \tag{18}$$

300 Therefore, it is not clear if we can directly apply the approach of [42] discussed in Section 2.2. Instead,
301 we have to use the linearization approach based on the high-order moment bounds for the remainder
302 term $D^{\mathsf{b}}$ (see Proposition 3 in Appendix E). This justifies the strong bounded noise assumption A2,
303 that we had to impose. We state the main result of this section below:

304 **Theorem 3.** *Assume A1 - A7. Then with $\mathbb{P}$ - probability at least $1 - 2/n$, it holds*

$$\sup_{B \in \mathsf{C}(\mathbb{R}^d)} |\mathbb{P}^{\mathsf{b}}(\sqrt{n}\{\Sigma_n^{\mathsf{b}}\}^{-1/2}(\bar{\theta}_n^{\mathsf{b}} - \bar{\theta}_n) \in B) - \mathbb{P}^{\mathsf{b}}(Y^{\mathsf{b}} \in B)| \leq \frac{M_{3,1}^{\mathsf{b}}}{n^{1/2}} + \frac{M_{3,2}^{\mathsf{b}} \log n}{n^{\gamma - 1/2}} + \frac{M_{3,3}^{\mathsf{b}} \log^{3/2} n}{n^{\gamma/2}} \, ,$$

305 *where $\{M_{3,i}^{\mathsf{b}}\}_{i=1}^3$ are defined in Appendix E.6, equation (61).*

306 *Proof sketch of Theorem 3.* We apply the bound

$$\sup_{B \in \mathsf{C}(\mathbb{R}^d)} |\mathbb{P}^{\mathsf{b}}(\{\Sigma_n^{\mathsf{b}}\}^{-\frac{1}{2}}(W^{\mathsf{b}} + D^{\mathsf{b}}) \in B) - \mathbb{P}^{\mathsf{b}}(Y^{\mathsf{b}} \in B)|$$

$$\leq \sup_{B \in \mathsf{C}(\mathbb{R}^d)} |\mathbb{P}^{\mathsf{b}}(\{\Sigma_n^{\mathsf{b}}\}^{-\frac{1}{2}} W^{\mathsf{b}} \in B) - \mathbb{P}^{\mathsf{b}}(Y^{\mathsf{b}} \in B)| + 2c_d (\mathbb{E}^{\mathsf{b}}[\|\{\Sigma_n^{\mathsf{b}}\}^{-\frac{1}{2}} D^{\mathsf{b}}\|^p])^{\frac{1}{1+p}} \, , \tag{19}$$

307 where $c_d \leq 4d^{1/4}$ is the isoperimetric constant of the class of convex sets. The proof of (19) is
308 provided in Proposition 3 in Appendix E. We can control $\bar{\mathbb{E}}[\|D^{\mathsf{b}}\|^p]$ by Burkholder's inequality, where
309 $\bar{\mathbb{E}}$ denotes the expectation w.r.t. the product measure $\mathbb{P}_\xi^{\otimes n} \otimes \mathbb{P}_w^{\otimes n}$. Then we proceed with Markov's
310 inequality to obtain $\mathbb{P}$ – high-probability bounds on the behavior of $\mathbb{E}^{\mathsf{b}}[\|D^{\mathsf{b}}\|^p]$. This result requires
311 us to provide bounds for

$$\bar{\mathbb{E}}^{1/p}[\|\theta_k - \theta^\star\|^p] \quad \text{and} \quad \bar{\mathbb{E}}^{1/p}[\|\theta_k^{\mathsf{b}} - \theta^\star\|^p] \, , \quad k \in \{1, \dots, n - 1\} \, , \tag{20}$$

312 with $p \simeq \log n$ and polynomial dependence on $p$. To control the second term in the r.h.s. of (19) we
313 note that the matrix $\Sigma_n^{\mathsf{b}}$ concentrates around $\Sigma_n$ due to the matrix Bernstein inequality (see Lemma 18
314 for details). Hence, there is a set $\Omega_1$ such that $\mathbb{P}(\Omega_1) \geq 1 - 1/n$ and $\lambda_{\min}(\Sigma_n^{\mathsf{b}}) > 0$ on $\Omega_1$. Moreover,
315 on this set we may use Berry-Essen-type bound for non-i.i.d. sums of random vectors. Detailed proof
316 is given in Appendix E. □

## 2.4 Rate of convergence in the Polyak–Juditsky central limit theorem

318 In the final part of this section, we discuss the issue of transition from $\Sigma_n$ to $\Sigma_\infty$ and estimation of
319 convergence rates in the Polyak–Juditsky result (4). We utilize the result of Theorem 2 together with
320 the following lemma.

321 **Lemma 1.** *Assume that A1 and A9 hold. Let $Y, Y' \sim \mathcal{N}(0, \mathrm{I}_d)$. Then, the Kolmogorov distance*
322 *between the distributions of $\Sigma_n^{1/2} Y$ and $\Sigma_\infty^{1/2} Y'$ is bounded by*

$$\mathsf{d}_{\mathsf{C}}(\Sigma_n^{1/2} Y, \Sigma_\infty^{1/2} Y') \leq C_\infty n^{\gamma - 1} \, ,$$

323 *where the constant $C_\infty$ is defined in (50).*

324 Theorem 2, Lemma 1, and triangle inequality imply the following result on closeness to $\mathcal{N}(0, \Sigma_\infty)$.

**Theorem 4.** *Assume A1, A3, A8(4), A9. Then, with $Y \sim \mathcal{N}(0, I_d)$ it holds that*

$$d_C(\sqrt{n}(\bar{\theta}_n - \theta^\star), \Sigma_\infty^{1/2} Y) \leq \frac{C_4}{\sqrt{n}} + \frac{C_5}{n^{\gamma - 1/2}} + \frac{C_6}{n^{\gamma/2}} + \frac{C_\infty}{n^{1-\gamma}} , \tag{21}$$

*where $C_4, C_5$ and $C_6$ are given in Theorem 2.*

**Discussion.** Theorem 2 reveals that the normal approximation through $\mathcal{N}(0, \Sigma_n)$ improves when the step sizes $\alpha_k$ are less aggressive, that is, as $\gamma \to 1$. However, Theorem 4 shows that there is a trade-off, since the rate at which $\Sigma_n$ converges to $\Sigma_\infty$ also affects the overall quality of the approximation. Optimizing the bound in (21) for $\gamma$ yields an optimal value of $\gamma = 3/4$, leading to the following approximation rate:

$$d_C(\sqrt{n}(\bar{\theta}_n - \theta^\star), \Sigma_\infty^{1/2} Y) \leq \frac{C'_1}{n^{1/4}} + \frac{C'_2}{\sqrt{n}}(\|\theta_0 - \theta^\star\| + \|\theta_0 - \theta^\star\|^2) ,$$

where $C'_1$ and $C'_2$ are instance-dependent quantities (but not depending on $\|\theta_0 - \theta^\star\|$), that can be inferred from Theorem 4. Given the result of Theorem 4 one can proceed with a non-asymptotic evaluation of the methods for constructing confidence intervals based on direct estimation of $\Sigma_\infty$, such as [11, 49].

**Lower bounds** We provide a lower bound indicating that the bound Theorem 4 is tight at least in some regimes of step size decay power $\gamma \in (1/2, 1)$. For this aim we consider minimization problem (1) of the following form:

$$f(\theta) = \theta^2/2 \to \min_{\theta \in \mathbb{R}} , \quad \theta_0 = 0 .$$

In this case $\theta^\star = 0$. We consider an additive noise model, that is, the stochastic gradient oracles $F(\theta, \xi)$ have a form $F(\theta, \xi) = \theta + \xi$, where $\xi \sim \mathcal{N}(0, 1)$. Enrolling (2), we get

$$\theta_k = -\sum_{j=1}^{k} \alpha_j \prod_{\ell = j+1}^{k} (1 - a\alpha_\ell)\xi_j \text{ and } \sqrt{n}\bar{\theta}_n = -\frac{1}{\sqrt{n}} \sum_{j=1}^{n-1} Q_j \xi_j , \tag{22}$$

where $Q_j = \alpha_j \sum_{k=j}^{n-1} \prod_{\ell=j+1}^{k} (1 - \alpha_\ell)$. Note that $\sqrt{n}(\bar{\theta}_n - \theta^\star)$ follows normal distribution $\mathcal{N}(0, \sigma_{n,\gamma}^2)$ with $\sigma_{n,\gamma}^2 = \frac{1}{n} \sum_{j=1}^{n-1} Q_j^2$. Due to Lemma 1 (see also equation (49) in the Appendix), we have $G = 1, \Sigma_\infty = 1$, and $\sigma_{n,\gamma}^2 \to 1$ as $n \to \infty$. Moreover, the following lower bound holds:

**Proposition 1.** *Consider the sequence $\{\theta_k\}_{k \in \mathbb{N}}$ defined by the recurrence (22) with $\alpha_j = c_0/(1+j)^\gamma$. Then it holds, for the number of observations $n$ sufficiently large, that*

$$|\sigma_{n,\gamma}^2 - 1| > \frac{C_1(\gamma, c_0)}{n^{1-\gamma}} , \tag{23}$$

*where the constant $C_1(\gamma, c_0)$ depends only upon $c_0$ and $\gamma$. Moreover, for $n$ large enough*

$$d_C(\sqrt{n}(\bar{\theta}_n - \theta^\star), \mathcal{N}(0, 1)) > \frac{C_2(\gamma, c_0)}{n^{1-\gamma}} . \tag{24}$$

**Discussion.** Proof of Proposition 1 is provided in Appendix F, together with some simple numerical simulations which indicate the tightness of the lower bound (23). Note that the bound (24) reveals that the distribution of $\sqrt{n}(\bar{\theta}_n - \theta^\star)$ can not be approximated by $\mathcal{N}(0, \Sigma_\infty)$ with the rate faster than $1/n^{1-\gamma}$. Moreover, it shows that the rate of normal approximation in Theorem 4 can not be improved when $\gamma \in [3/4; 1)$. This fact is extremely important when taking into account the bootstrap validity result of Theorem 1 and normal approximation in Theorem 2. Indeed, both results suggests that the rates of normal approximation of order up to $1/\sqrt{n}$ can be achieved when $\gamma \to 1$, but they require to consider another covariance matrix $\Sigma_n$, corresponding to the linearized recurrence in (13). At the same time, in the regime $\gamma \to 1$, the approximation by $\mathcal{N}(0, \Sigma_\infty)$ can be too slow. It is an interesting and, to the best of our knowledge, open question to provide lower bounds analogous to Proposition 1 which show the tightness of other summands in Theorem 4 in the regime $1/2 < \gamma < 3/4$.

## 3 Conclusion

In our paper, we performed the fully non-asymptotic analysis of the multiplier bootstrap procedure for SGD applied to strongly convex minimization problems. We showed that the algorithm can achieve approximation rates in convex distances of order up to $1/\sqrt{n}$. We highlight the fact that the validity of the multiplier bootstrap procedure does not require one to consider Berry-Esseen bounds with the asymptotic covariance matrix $\Sigma_\infty$, which is in sharp contrast to the methods that require direct estimation of $\Sigma_\infty$.

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
