# OpenReview forum: "Gaussian Approximation and Multiplier Bootstrap for Stochastic Gradient Descent"
_NeurIPS.cc/2025/Conference — Submitted to NeurIPS 2025_

### Official Review · Reviewer_iTNL · 2025-06-28

**Clarity:** 3
**Significance:** 2
**Originality:** 2
**Rating:** 4
**Confidence:** 4

**Summary:**

In this manuscript, the authors investigate the limit theorems for Noisy SGD
$$
\theta_{k} = \theta_{k} -\alpha_{k}(\nabla f(\theta_{k-1}) + \xi_{k}).
$$
The sequence of averaged iterates is defined as $\sqrt{n}(\bar{\theta}_{n} - \theta^{\ast})$.

Under some assumptions, the authors obtain the convergence rate of the averaged iterates to a zero-mean Gaussian distribution with covariance matrix $\Sigma_{\infty}$ in terms of convex distance $\mathrm{d}_{\mathrm{C}}(\cdot,\cdot)$.

**Questions:**

The questions that I am concerned about have been listed in the 'Weaknesses'.

**Ethical Concerns:**

["NO or VERY MINOR ethics concerns only"]

**Final Justification:**

I keep my rating.

**Limitations:**

Yes

**Paper Formatting Concerns:**

The presentation of this manuscript satisfied the formatting requirements of NeurIPS 2025.

**Quality:**

3

**Strengths And Weaknesses:**

Strengths:

1. The author provide the lower bounds on the convex distance $\mathrm{d}_{\mathrm{C}}(\sqrt{n}(\bar{\theta}_n - \theta^{\ast}),\mathcal{N}(0,\Sigma))$.
2. It is interesting to note that the multiplier bootstrap procedure is employed to study the convergence result of SGD.


Weaknesses:

1. Since the authors assume the decomposition $F(\theta_{k-1}, \xi_k) = \nabla f(\theta_{k-1}) + \xi_{k}$ holds, they actually study the noisy gradient descent (7) rather than the general SGD (2). I suggest the author should clarify this aspect in advance.
2. As we know $\nabla f(\theta)=\mathbb{E}_{\xi\sim\mathbb{P}}[F(\theta, \xi)]$, I wonder whether these assumptions (A1)-(A4) are mutually compatible, as their compatibility is crucial for the validity of the results. I would appreciate it if you could give some explanation.
3. Could you please give an example of the objective function that satisfies A1-A4?
4. In the following references, the authors have considered the limit theorems for SGD, the results in which are closed to this manuscript. Could you please provide a detailed comparison between these literatures and your works? I believe this would help us better understand the contributions of your research.

    [1] Jose Blanchet, Aleksandar Mijatović, and Wenhao Yang. Limit Theorems for Stochastic Gradient Descent with Infinite Variance. arXiv:2410.16340v3, 2024.

    [2] Jerome Sacks. Asymptotic distribution of stochastic approximation procedures. The Annals of Mathematical Statistics, 29(2):373–405, 1958.
5. I would be glad to see more validation of the theoretical result. So could you please provide some numerical experiments to illustrate their theoretical results?

---

> ### Author Rebuttal · Authors · 2025-07-31
>
> We thank the reviewer for their feedback and for acknowledging the contributions of our paper. We would like to address further the reviewer's concerns about the paper's novelty and main strengths and weaknesses of the submission.
>
> ***It is interesting to note that the multiplier bootstrap procedure is employed to study the convergence result of SGD.***
> We would like to clarify that the multiplier bootstrap procedure is **not** employed to study the convergence properties of SGD itself. Rather, it is used to construct confidence sets for the averaged SGD iterate $\bar{\theta}_n$. Our theoretical contribution lies in establishing non-asymptotic validity of this procedure.
>
> Q1. ***The noisy gradient setting*** We would like to clarify that we actually consider the general SGD algorithm (2), not just the special case of noisy gradient descent. Specifically, we ***do not assume*** that the noise term $\zeta_k$ in the decomposition
> $$
> F(\theta_{k-1}, \xi_k) = \nabla f(\theta_{k-1}) + \zeta_k
> $$
> is independent of $\theta_{k-1}$. Contrary, as stated in Assumptions A2 and A8, we assume that
> $$
> \zeta_k = g(\theta_{k-1}, \xi_k) + \eta(\xi_k),
> $$
> where $g(\theta_{k-1}, \xi_k)$ is a martingale difference term and \eta(\xi_k) is independent of $\theta_{k-1}$. This structure is standard in the analysis of general stochastic gradient methods. One can rewrite the standard SGD setting with the access to stochastic oracles in this way. Namely, given an access to the function $F(\theta, \xi)$ taking values in $\mathbb{R}^d$, such that
> $$
> \mathbb{E}_\xi [F(\theta, \xi)] = \nabla f(\theta),
> $$
> we can set $\eta(\xi) =  F(\theta^{\star}, \xi)$ and $g(\theta, \xi) = F(\theta,\xi)- F(\theta^{\star}, \xi) - \nabla f(\theta)$. This discussion is provided in the main text of the paper, lines $148-156$. In the revised version of the paper we will revise the text to further clarify this point.
>
> Q2. We agree with the reviewer that the imposed assumptions are rather restrictive. At the same time, they can be partially relaxed. For the detailed answer please see our response to the weakness 1 (W1) raised by the Reviewer ***x2pB***. We briefly mentions the key points here. In particular, we can remove the coercivity assumption A4. The main reason for including the the coercivity assumption was to simplify the proof of the high-probability bounds for $\|\theta_n - \theta^*\|$. It is not essential — the result still holds without it, although the constants in the proof would change slightly. In the revised version of the paper we will remove this assumption and clarify that it is not essential for the main result.
>
> Assumptions A1 and A4 are often considered when analyzing SGD algorithms, see [5,6]. Assumption A2 is not standard only in terms of the noise boundedness assumption A2(iii). As we pointed out in the paper, assumption A2(iii) is used only to prove the high-probability bound for the last iterate in the "bootstrap world", and the result about normal approximation in the "real" world (Theorem 2) holds without this assumption. In the latter case, we use assumption A8, which is considerably milder. At the same time, assumption A2(iii) can be partially relaxed as outlined below. Recall that the noise component $\zeta_k$ consists of two parts:
> $$
> \zeta_k = g(\theta_k, \xi_{k+1}) + \eta(\xi_{k+1})
> $$
> We can allow the part $\eta(\xi_{k+1})$ to be unbounded (e.g. sub-Gaussian or just admitting a finite exponential moment). Our current proof technique also allows us to consider unbounded component $g(\theta_k, \xi_{k+1})$, but in this case we have to assume that a random variable $g(\theta, \xi_1)$ has a sub-Gaussian parameter that does not depend upon $\theta$. This limitation comes from the high-probability bounds for the last iterate, which are, due to our knowledge, unavoidable, see detailed discussion in  the paper [4] and in lines 157-175 in current submission.
>
> Q3. ***Could you please give an example of the objective function that satisfies A1-A4?*** A simple particular example of such a function is as follows:
> $$
> f(x) = x^2 + cos(x)\,,
> $$
> where the noise oracle is given by
> $$
> F(x,\xi) = 2x-sin(x)*\xi
> $$
> with the noise variable $\xi \sim U[0.5, 1.5]$. A general pipeline for constructing such functions is to consider a bounded perturbation of the quadratic objective by some additional component, which has bounded derivatives. Of course, this is not the only setting covered in our paper, but rather a particular example.
>
> Q4. ***Related references*** We thank the reviewer for pointing out these related works. Both [1] and [2] focus on the asymptotic analysis of stochastic gradient descent and study the limiting distribution of the last iterate. In contrast, in our work we consider the non-asymptotic analysis of the Polyak-Ruppert averaged iterates, not the last iterate, and on constructing confidence intervals for this estimator. We additionally highlight that our analysis is fully non-asymptotic.
>
> Q5. ***Empirical Illustration.*** While we analyze theoretically the multiplier bootstrap algorithm, we are not the ones who suggested this algorithm. It was proposed in the paper [3], where the authors performed empirical validation of their approach compared to the methods which uses the plug-in and batch-mean approaches to estimate the asymptotic covariance matrix. That is why we did not include an empirical illustration comparing the multiplicative bootstrap with alternative approaches. We can revise the final version of the paper to include an experiment to compare the underlying methods in terms of coverage probabilities, at the same time, we can not obtain these results before the end of rebuttal period.
>
>
> ***References***
>
> [1] Jose Blanchet, Aleksandar Mijatović, and Wenhao Yang. Limit Theorems for Stochastic Gradient Descent with Infinite Variance. arXiv:2410.16340v3, 2024.
>
> [2] Jerome Sacks. Asymptotic distribution of stochastic approximation procedures. The Annals of Mathematical Statistics, 29(2):373–405, 1958.
>
> [3] Yixin Fang, Jinfeng Xu, and Lei Yang. Online bootstrap confidence intervals for the stochastic gradient descent estimator. Journal of Machine Learning Research, 19(78):1–21, 2018.
>
> [4] Nicholas JA Harvey, Christopher Liaw, Yaniv Plan, and Sikander Randhawa. Tight analyses for non-smooth stochastic gradient descent. In Conference on Learning Theory, pages 1579–1613. PMLR, 2019.
>
> [5] Andreas Anastasiou, Krishnakumar Balasubramanian, and Murat A. Erdogdu. Normal approximation for stochastic gradient descent via non-asymptotic rates of martingale CLT. In Alina Beygelzimer and Daniel Hsu, editors, Proceedings of the Thirty-Second Conference on Learning Theory, volume 99 of Proceedings of Machine Learning Research, pages 115–137. PMLR, 25–28 Jun 2019.
>
> [6] Qi-Man Shao and Zhuo-Song Zhang. Berry–Esseen bounds for multivariate nonlinear statistics with applications to M-estimators and stochastic gradient descent algorithms. Bernoulli, 28(3):1548–1576, 2022.

---

> > ### Comment · Reviewer_iTNL · 2025-08-03
> >
> > Thank you very much for the response! I think the authors have addressed all my concerns. As long as the authors incorporate all the promised changes one by one in the final submission, I would be happy to see this paper at NeurIPS.

---

### Official Review · Reviewer_tXDr · 2025-07-02

**Clarity:** 4
**Significance:** 3
**Originality:** 3
**Rating:** 4
**Confidence:** 2

**Summary:**

The authors study the properties of a bootstrap approximation for Polyak-Ruppert SGD iterates.  For strongly convex optimization problems, they establish a rate of convergence in the convex distance of $1/\sqrt{n}$, matching the Berry-Esseen rate.  To prove this result, the authors show that the bootstrap is able to approximate an intermediate covariance matrix that differs from the Polyak-Juditsky covariance matrix.

**Questions:**

- While the authors establish a sharper rate of convergence for the bootstrap, the technical conditions are stronger.  It would be appreciated if the authors could discuss if there are any substantial differences in the conditions compared to plug-in approaches to inference.
- In their discussion of the proof of Theorem 1, the authors provide an expression for $\Sigma_n$.  It would be helpful if the authors briefly discuss why plug-in estimators for this quantity cannot readily be constructed.
-  The paper of Shao and Zhang (2022) also considers SGD as an application of their Berry-Esseen bound for non-linear statistics.  It would be helpful if the authors could explain the differences of the proof of Theorem 2 from what is considered in this paper.

**Ethical Concerns:**

["NO or VERY MINOR ethics concerns only"]

**Final Justification:**

I am maintaining my original positive score.

**Limitations:**

Yes

**Quality:**

4

**Strengths And Weaknesses:**

Strengths:
- The authors establish a sharp rate of convergence for the bootstrap approximation, improving on other plug-in approaches in the literature.
- The observation that the bootstrap offers a more direct approximation of the sampling distribution, leading to faster rates of convergence than methods that target the asymptotic distribution, may be of interest in other related problems.
- The paper is well-written overall.

Weaknesses:
- It appears that some additional regularity conditions that are not needed for the CLT are required to study the bootstrap.  However, the authors do a good job of explaining the conditions and citing examples where similar assumptions are made.
- While the results are sharp, it appears that the result holds only for $d$ fixed and for very well-behaved optimization problems.

---

> ### Author Rebuttal · Authors · 2025-07-31
>
> We thank the reviewer for their positive feedback and are happy to provide further details in response to their concerns. We are pleased that the reviewer acknowledged that our paper is well-written and provides novel results on the comparison of bootstrap-based methods and the ones based on plug-in the the context of SGD. Below we provide our responses to the reviewer's questions (Q) and weaknesses (W).
>
> W1. ***It appears that some additional regularity conditions that are not needed for the CLT are required to study the bootstrap*** We agree that some of our assumptions might be restrictive. Some of them can be partially relaxed, for the detailed answer please see our response to the weakness 1 (W1) raised by the Reviewer ***x2pB***.
>
> Q1. ***While the authors establish a sharper rate of convergence for the bootstrap, the technical conditions are stronger. It would be appreciated if the authors could discuss if there are any substantial differences in the conditions compared to plug-in approaches to inference.*** This is a very important question. We would like to clarify that, to the best of our knowledge, there is no fully non-asymptotic analysis of the batch-mean or plug-in based methods for in the context of SGD. Typically the authors provide only in-expectation bounds, that is, they suggest some estimator $\hat\Sigma_{n}$ of the asymptotic variance $\Sigma_{\infty}$ and provide non-asymptotic bounds for the quantities, such as
> $$
> \mathbb{E}\|\hat\Sigma_n - \Sigma_{\infty}\|,
> $$
> and do not establish high-probability guarantees. Such in-expectation bounds are not sufficient for our purposes. In particular, they cannot be used to prove a Gaussian comparison result  for distributions $\mathcal{N}(0, \hat\Sigma_n)$ and $\mathcal{N}(0, \Sigma_\infty)$ on sets with high $\mathbb{P}$-probability. Moreover, the papers, which study the non-asymptotic properties of the batch-mean and plug-in estimators, typically do not prove the non-asymptotic CLT and rely, at the end of the day, only on the ***asymptotic*** validity of the derived confidence intervals. In addition, plug-in approaches for SGD typically rely on access to second-order stochastic oracles (that is, the stochastic Hessian matrix) to estimate \$\Sigma\_\infty\$, which is not required in our approach. In summary, our stronger assumptions are due to the fact that we carry out a complete analysis of the procedure, contrary to the in-expectation bounds on the approximation accuracy of the limiting covariance matrix. To the best of our knowledge, obtaining similar non-asymptotic guarantees for plug-in and batch-mean methods in context of SGD remains an open problem.
>
> Q2. ***It would be helpful if the authors briefly discuss why plug-in estimators for $\Sigma_n$ cannot readily be constructed*** Constructing a plug-in estimator for the linearized covariance matrix $\Sigma_n$ is difficult. Indeed, recall that
> $$
> \Sigma_n =  n^{-1} \sum_{k=1}^{n-1} Q_k \Sigma_{\varepsilon} Q_k^{\top}
> $$
> where the matrices $Q_i$ have a form
> $$
> Q_i = \alpha_i\sum_{j=i}^{n-1}\prod_{k=i+1}^{j}(I-\alpha_k G)\,,
> $$
> and $G = \nabla^2 f(\theta^{\star})$ is Hessian at optimal point. First, from this construction it is clear that estimating $\Sigma_n$ requires to estimate the Hessian matrix $G$, which in turn requires access to a second-order stochastic oracle, which we do not assume. Second, even if such an estimator was available, one would still need to derive bound for Gaussian comparison for distributions $\mathcal{N}(0, \hat{\Sigma}_n)$ and $\mathcal{N}(0, \Sigma_n)$, which is a nontrivial task, which can introduce additional error terms. So, in principal, $\Sigma_2$ can be estimated, but
>
> a. It requires a second-order stochastic oracle;
>
> b. The corresponding procedure would require a separate analysis and it is unclear that it is going to improve over the multiplier bootstrap rates.
>
> It is an interesting direction to study this approach and compare the actual rates with the ones arising from the bootstrap procedure. We will incorporate this discussion to the revised version of the paper.
>
> Q3. ***It would be helpful if the authors could explain the differences of the results of (Shao and Zhang, 2022) from what is considered in Theorem 2 in this submission.***  We would like to clarify that Theorem 2 is not a primary contribution of our work.  It plays an auxiliary role in establishing the  rate of convergence in the Polyak-Juditsky central limit theorem (Theorem 4), and provides a basis for the bootstrap validity part in Theorem 1. At the same time, our proof of normal approximation in the bootstrap world (Theorem 3), which is more involved technically, does not follow the pipeline of Shao and Zhang, since it essentially relies on the high-probability bounds on the remainder statistics $D^{b}$ with respect to the bootstrap-world distribution $\mathbb{P}^{b}$. As emphasized in the paper (see discussions in lines $293-300$), there are substantial technical differences that prevents us from using this approach here.  While the proof of Theorem 2 follows the general approach of Shao and Zhang (2022), we improve it by making all constants explicit, which were omitted in their paper. We also rely on this result when proving lower bounds on the accuracy of Gaussian approximation with $\Sigma_{\infty}$ stated in Proposition 1.
>
> W2.***It appears that the result holds only for $d$ fixed.*** We agree that our theoretical results are established under the assumption that the parameter dimension $d$ is fixed. However, we would like to note that it is not immediately clear what the appropriate formulation of the stochastic gradient descent algorithm should be in the setting where the dimension $d$ of the parameter $\theta$ grows with the number of iterations. We would greatly appreciate it if the reviewer could clarify this  comment.

---

> > ### Comment · Reviewer_tXDr · 2025-08-04
> >
> > I thank the authors for the detailed rebuttal.  I am also maintaining my positive score.

---

### Official Review · Reviewer_x2pB · 2025-07-02

**Clarity:** 3
**Significance:** 3
**Originality:** 3
**Rating:** 4
**Confidence:** 2

**Summary:**

This paper addresses the problem of constructing statistically valid confidence sets for the optimum $\theta^$ of a strongly convex objective when using stochastic gradient descent (SGD). The authors analyze a multiplier bootstrap procedure for Polyak–Ruppert averaged SGD iterates, which involves generating perturbed SGD trajectories to approximate the distribution of $\sqrt{n}(\bar{\theta}_n - \theta^)$. They prove, under suitable smoothness and noise conditions, that the bootstrap can accurately approximate the quantiles of the exact error distribution at finite sample sizes. Notably, their non-asymptotic bounds show an approximation error of order $n^{-\xi/2}$ (up to log factors) for step-size schedules $\alpha_k\propto k^{-\xi}$ with $\xi\in(0.5,1)$. This provides the first rigorous finite-sample guarantee for bootstrap-based confidence intervals in SGD. Furthermore, the analysis reveals that the bootstrap’s validity hinges on approximating a finite-$n$ covariance ($\Sigma_n$) rather than the usual asymptotic covariance $\Sigma_1$, avoiding the need to directly estimate $\Sigma_1$. In addition, the paper establishes new rates for the convergence of the averaged SGD iterate’s distribution to normality (the Polyak–Juditsky CLT): the distributional distance to $N(0,\Sigma_1)$ is shown to be $O(n^{-1/4})$ for a specific step-size schedule, and a matching lower bound is provided to demonstrate the tightness of this rate. Overall, the contributions advance the theory of uncertainty quantification for SGD by providing non-asymptotic bootstrap confidence guarantees and deeper insight into the distributional behavior of SGD iterates.

**Questions:**

1. Relaxing the Noise Assumption: One major assumption is the almost sure boundedness of the stochastic gradient noise (Assumption A2(iii) and related conditions). How sensitive are the results to this assumption? Could the analysis be extended to allow sub-Gaussian noise or heavier-tailed noise (perhaps with slower rates or additional logarithmic factors)?

2. Empirical Illustration: To increase the practical impact, consider adding a small-scale experiment or simulation. For example, run SGD on a simple logistic regression or quadratic problem and construct confidence intervals using the multiplier bootstrap vs. using a plug-in covariance estimate.

**Ethical Concerns:**

["NO or VERY MINOR ethics concerns only"]

**Final Justification:**

The authors have done an adequate job addressing my concerns therefore I maintain my initial positive rating of 4.

**Limitations:**

Yes

**Quality:**

3

**Strengths And Weaknesses:**

**Strengths**
1. The paper presents a thorough theoretical analysis with clear assumptions (A1–A9) and complete proofs (deferred to appendices). The main theorems are stated with high-probability guarantees and rates, and the authors even derive a lower bound to confirm that their key upper bound is tight.
2. The contributions advance the theoretical state-of-the-art in several ways. Establishing finite-sample convergence rates up to $O(1/\sqrt{n})$ for distributional approximations is non-trivial and improves upon earlier results that were either asymptotic or slower. This is the first work to show that a bootstrap method can achieve a faster distributional convergence rate than the classic CLT rate for SGD (which, in terms of Berry–Esseen bounds for $\sqrt{n}(\bar{\theta}_n-\theta^*)$, was effectively slower).
3. The presentation of the paper is clear with a nice flow.

**Weaknesses**

1. The main limitations in quality stem from the restrictive conditions required. Notably, the analysis assumes the stochastic gradient noise is essentially bounded (almost sure boundedness of $\zeta_k$ in Assumption A2(iii)) and that the stochastic gradient is co-coercive (Assumption A4). These conditions are quite strong and may not hold in many practical scenarios.

2. The paper’s novelty is incremental in the sense that it builds directly on an existing methodology (the bootstrap algorithm from [16]) and extends recent theoretical frameworks. The work is a natural next step after [38] and [46], and some techniques are adapted from those sources (as the authors acknowledge, e.g. using techniques from [42] and [46] in proving their Theorem 2). Thus, the contribution is not a brand-new algorithm or completely unconventional theory, but rather a meaningful extension and generalization of prior art.

---

> ### Author Rebuttal · Authors · 2025-07-31
>
> We thank the reviewer for their positive feedback and are happy to provide further details in response to their concerns. We are pleased that the reviewer acknowledged that our paper is well-written and provides novel theoretical contributions. Below we provide our responses to the reviewer's questions (Q) and weaknesses (W).
>
> W1 & Q1: ***The main limitations in quality stem from the restrictive conditions required.*** & ***Relaxing the Noise Assumptions***
>
> We agree with the reviewer that the assumptions A2(iii) gradient's co-coercivity (Assumption A4) can be rather restrictive. They can be partially relaxed as discussed below. First, we can actually remove the coercivity assumption A4 (and there is a small note about it in the paper).  The main reason for including this assumption was to simplify the proof of the high-probability bounds for $\|\theta_n - \theta^*\|$. It is not essential — the result still holds without it, although the constants in the proof would change slightly. In the revised version of the paper we will remove this assumption and clarify that it is not essential for the main result.
>
> The situation about the boundedness assumption A2(iii) on the noise in the stochastic gradient is more delicate. This assumption is used only to prove the high-probability bound for the last iterate (both in the "real world" and in the "bootstrap world"), and the result about normal approximation in the "real" world (Theorem 2) holds without this assumption: we require only the assumption A8(4), which is considerably weaker. Unfortunately, existing high-probability results on the last iterate of SGD allows only a partial relaxation of A2(iii) as outlined below. Recall that the noise component $\zeta_k$ consists of two parts:
> $$
> \zeta_k = g(\theta_k, \xi_{k+1}) + \eta(\xi_{k+1})
> $$
> We can allow the additive component of the noise $\eta(\xi_{k+1})$ to be unbounded (e.g. sub-Gaussian or just admitting a finite exponential moment). This will not require almost any modifications in the proofs. Our current proof technique also allows us to consider unbounded component $g(\theta_k, \xi_{k+1})$, but in this case we have to assume that a random variable $g(\theta, \xi_1)$ has a sub-Gaussian parameter that does not depend upon $\theta$. Effectively this can be achieved only in the setting when $g(\theta, \xi_1)$ is bounded, yet such a generalization is possible. Otherwise, as already stated, our analysis requires a technique to obtain the high-probability error bounds for the last iterate of SGD, that is, $\|\theta_n - \theta^{\star}\|$. If it is done under milder assumptions compared to the ones stated above, our analysis readily improves.
>
> W2. ***The paper’s novelty is incremental in the sense that it builds directly on an existing methodology***
> While our work builds on existing techniques, it introduces several novel and technically nontrivial contributions that go beyond prior literature. The main novelty lies in establishing the non-asymptotic validity of the multiplier bootstrap procedure when using $\Sigma_n$ in the Gaussian approximation. In particular, we obtain convergence rates up to $n^{-1/2}$ as $\gamma \to 1$, which improves upon the results in \[38, 46] that consider similar procedures in the context of linear stochastic approximation and TD learning, respectively. The key reason for this improvement is essentially the fact that we proceed through the Gaussian approximation with the covariance matrix of the linearized autoregressive process $\Sigma_n$, and not through the asymptotic covariance matrix $\Sigma_{\infty}$. Moreover, we provide a lower bound that demonstrates that choosing $\Sigma_\infty$ for Gaussian approximation in the real world leads to a strictly worse rate for $\gamma \in (3/4, 1)$. This reveals an important feature related with the multiplier bootstrap approach: it essentially proceed in a different manner rather than approaching $\Sigma_{\infty}$, and our convergence rates and lower bounds support this claim.
>
> While our proof builds on the linearization in [42], both in Theorem 1 and 3, [42] operates only with the 4-th moment bound, while for our bootstrap validity proof it is crucial to control the $p$th moments of the SGD error $\theta_k - \theta^{\star}$ as well as its bootstrap counterpart $\theta_k^{b} - \theta^{\star}$ and to choose $p \simeq \log{n}$. The latter requires obtaining the error bounds with explicit scaling in $p$, which is not trivial, especially when working with the bootstrap iterates $\theta_k^{b} - \theta^{\star}$. In fact, the classical proofs in the stochastic approximation require that the step sizes $\alpha_k$ are monotonically decreasing, while this is no longer the case when analyzing $\theta_k^{b} - \theta^{\star}$. We managed to address this issue during the proof of Theorem 3.
>
> Q2. ***Empirical Illustration:*** While we analyze theoretically the multiplier bootstrap algorithm, we are not the ones who suggested this algorithm. It was proposed in the paper [16], where the authors performed empirical validation of their approach compared to the methods which uses the plug-in and batch-mean approaches to estimate the asymptotic covariance matrix. That is why we did not include an empirical illustration comparing the multiplicative bootstrap with alternative approaches. We can revise the final version of the paper to include an experiment to compare the underlying methods in terms of coverage probabilities, at the same time, we can not obtain these results before the end of rebuttal period.
>
> ***References***
>
> [16] Yixin Fang, Jinfeng Xu, and Lei Yang. Online bootstrap confidence intervals for the stochastic gradient descent estimator. Journal of Machine Learning Research, 19(78):1–21, 2018.
>
> [38] Sergey Samsonov, Eric Moulines, Qi-Man Shao, Zhuo-Song Zhang, and Alexey Naumov. Gaussian Approximation and Multiplier Bootstrap for Polyak-Ruppert Averaged Linear Stochastic Approximation with Applications to TD Learning. In Advances in Neural Information Processing Systems, volume 37, pages 12408–12460. Curran Associates, Inc., 2024.
>
> [42] Qi-Man Shao and Zhuo-Song Zhang. Berry–Esseen bounds for multivariate nonlinear statistics with applications to M-estimators and stochastic gradient descent algorithms. Bernoulli, 28(3):1548–1576, 2022.
>
> [46] Weichen Wu, Gen Li, Yuting Wei, and Alessandro Rinaldo. Statistical Inference for Temporal Difference Learning with Linear Function Approximation. arXiv preprint arXiv:2410.16106, 2024.

---

> > ### Comment · Reviewer_x2pB · 2025-08-04
> >
> > I thank the authors for the detailed response. I will maintain my positive score.

---

### Official Review · Reviewer_mjJN · 2025-07-03

**Clarity:** 3
**Significance:** 2
**Originality:** 2
**Rating:** 4
**Confidence:** 2

**Summary:**

This paper studies the rate of convergence of the average of the iterates from time zero to n called the Polyak--Ruppert averaging. It is well known that the fluctuations of this estimatoir are Gaussian, and the paper provides some finite $n$ bounds on this rate of convergence. The paper also studies the bootstrap proceduce, which does not depend on the limiting covariance $\Sigma_\infty$.

**Questions:**

1. The convex distance is a bit strange to me. Why is this distance used instead of a more standard ones like the Kolmogorov distance or the total variation distance. Does this metric have some nice properties that differ from these more standard ones?

2. Why do we study the bootstrap method? What are the main barriers from proving a rate of convergence directly on the usual SGD iterates.

3. How necessary are the assumption in this paper? In particular, if these assumptions where to be weakened, will the fluctuations no longer be Gaussian?

4. The importance of the role of $\Sigma_n$ and $\Sigma_\infty$ raised in point 2 on line 63 is a bit unclear. There should be concentration of the empirical average, so the error introduced from the approximation should be controlled. It is not clear that going through the bootstrap gives a more precise rate on convergence.

**Ethical Concerns:**

["NO or VERY MINOR ethics concerns only"]

**Final Justification:**

I am unfamiliar with this line of work. My questions and misunderstandings regarding the relevance of the assumptions and the importance of the analysis have been clarified by reading both the author responses and the other reviews. I have raised my score to be in line with the other reviewers.

**Limitations:**

Yes

**Paper Formatting Concerns:**

No issues

**Quality:**

2

**Strengths And Weaknesses:**

The problem of the rate of convergence studied in this paper is very natural and the paper provides a nice precise derivation of this rate of convergence by approximating the iterates with the iterates of a perturbed SGD. It is not surprising that such rates of convergence can be proved, but the analysis seems to be carefully done.

However, I have a few concerns that the scope of this paper might be quite narrow. Namely the convexity and coercivity assumptions seem to be quite restrictive. Rigorous bounds even under some strong assumptions on the objective are welcome, but perhaps its scope is too narrow for a venue like NEURIPs. The usage of the bootstrap procedure does not seem well motivated in this paper, and for the purpose of proving a rate a convergence does not seem like a necessary intermediate step.

---

> ### Author Rebuttal · Authors · 2025-07-31
>
> We thank the reviewer for their feedback. Below we address some points raised as weaknesses and provide responses to the listed questions.
>
> We first address point raised as a weaknesses:
> (Weakness 2)***The usage of the bootstrap procedure does not seem well motivated in this paper, and for the purpose of proving a rate a convergence does not seem like a necessary intermediate step.*** & (Question 4) ***Why do we study the bootstrap method?***
> The reviewer incorrectly assumes that the bootstrap argument is required to prove the Gaussian approximation limits for
> $\sqrt{n}(\bar{\theta_n} - \theta^\star)$. This is not the case. Conversely, the latter Gaussian approximation is instrumental to prove the bootstrap validity. Gaussian approximation rates and some lower bounds provided in the paper are ***not*** derived through the analysis of the perturbed SGD.
> Thus the main object of study in this paper is the bootstrap approximation for the original distribution of the SGD iterates. The importance of bootstrap is basically outlined in the paper's introduction: this is one of the two most popular approaches for constructing confidence intervals for $\theta^*$ alongside with the methods that try to estimate the asymptotic covariance matrix $\Sigma_\infty$ directly. The multiplicative bootstrap we study is particularly attractive because it is an **online procedure** that constructs confidence sets without requiring resampling from the data or generating new observations.
>
> Weakness 1: ***Namely the convexity and coercivity assumptions seem to be quite restrictive***
>
> a.) ***Convexity*** is necessary in order that the minimization problem $f(\theta) \to \min$ has a unique minimizer. Non-convex problems, with the local minimum, can be potentially studied with local linearization with the analysis essentially similar to the one used "globally" in the convex case, see e.g. [47]. At the same time, such a generalization adds additional level of technical complexity and is beyond the scope of the paper. We will discuss this reference in more details in the revised version of the paper.
>
> b.) ***Coercivity*** : as mentioned in the discussion in the main paper, the coercivity assumption can be removed. The main reason for including this assumption was to simplify the proof of the high-probability bounds for the last iterate error $\|\theta_n - \theta^*\|$. It is not essential — the result still holds without it, although the constants in the proof would change slightly. We will revise the final version of the paper to remove this assumption and clarify that it is not essential for the main result.
>
> Question 1: ***The convex distance is a bit strange to me*** It is well-known that, for discrete distributions, the total variation distance to any continuous limit is always equal to one. This makes it unsuitable for studying convergence in the CLT without assuming a density component in the noise. In contrast, convex distance is actually a natural generalization of the 1-dimensional Kolmogorov distance, as it gives control for the probabilities of hitting balls, rectangles, ellipsoids, or more sophisticated convex sets. This is particularly relevant in statistical applications, because confidence sets (e.g., confidence balls or ellipsoids) often have convex geometry and are naturally defined as ellipsoids in appropriately chosen norms. Thus, convergence in convex distance directly translates into guarantees for the accuracy of confidence regions, which is our main motivation for using it in the analysis.
>
> Question 2: ***How necessary are the assumptions ....*** The assumptions required for the Gaussian approximation in the "real world" are standard and closely follow those used in the classical Polyak–Juditsky CLT [31]. Similar assumptions also appear in recent works such as [42]. We highlight that these assumptions are imposed in the original Polyak-Juditsky paper even for the asymptotic result to hold. The additional assumptions for the Gaussian approximation in the "bootstrap world" primarily arise from the high-probability bounds for the last SGD iterate. These bounds are instrumental for the non-asymptotic validity of the bootstrap procedure. Regarding Assumption A4, please refer to our response under point 1. The situation about the boundedness assumption on the noise in the stochastic gradient is more delicate. This assumption is used only to prove the high-probability bound for the last iterate, and the result about normal approximation in the "real" world (Theorem 2) holds without this assumption. At the same time, it is possible to partially relax this assumption as outlined below. Recall that the noise component $\zeta_k$ consists of two parts: $\zeta_k = g(\theta_k, \xi_{k+1}) + \eta(\xi_{k+1})$. We can allow the part $\eta(\xi_{k+1})$ to be unbounded (e.g. sub-Gaussian or just admitting a finite exponential moment). Our current proof technique also allows us to consider unbounded component $g(\theta_k, \xi_{k+1})$, but in this case we have to assume that a random variable $g(\theta, \xi_1)$ has a sub-Gaussian parameter that does not depend upon $\theta$. Effectively this can be achieved only in the setting when $g(\theta, \xi_1)$ is bounded.
>
> ***The importance of the role of $\Sigma_n$ and $\Sigma_\infty$ raised in point 2 on line 63 is a bit unclear. There should be concentration of the empirical average, so the error introduced from the approximation should be controlled. It is not clear that going through the bootstrap gives a more precise rate on convergence.*** We would like to emphasize again that we are ***not*** going through the bootstrap world to prove the result for original SGD error, see reply to the Weakness 2. From a high-level perspective, $\Sigma_n$ is important, because the rates of Gaussian approximation with $\Sigma_n$ are faster than the ones with $\Sigma_\infty$. If all the analysis is done in terms of $\Sigma_{\infty}$ directly, it is most likely that the final convergence rates of the bootstrap procedure are not better compared to those corresponding to the plug-in or batch-mean methods, which aim to estimate the covariance matrix $\Sigma_{\infty}$ directly. The analysis, carried out in terms of $\Sigma_n$, shows that multiplier bootstrap procedure analyzed in the paper is doing something different rather than just approximating $\mathcal{N}(0,\Sigma_{\infty})$, and this is supported by our final rates of convergence, which approaches $1/\sqrt{n}$.
> To acknowledge the importance of $\Sigma_n$ for the proof of ***bootstrap approximation***, we refer to the scheme of the proof we employed. The proof proceeds in three key steps (see the proof sketch of Theorem 1 in Section 2.1):
>
> a. We first derive the rate of convergence of the averaged SGD iterate $\sqrt{n}(\bar{\theta}_n - \theta^*)$ to the Gaussian distribution $\mathcal{N}(0, \Sigma)$ in the real world.
> b. We then show that the bootstrap analogue converges to $\mathcal{N}(0, \Sigma^b)$ in the bootstrap world.
> c. Finally, we carry out a Gaussian comparison argument between $\mathcal{N}(0, \Sigma)$ and $\mathcal{N}(0, \Sigma^b)$.
>
> An important question here is the choice of the approximating Gaussian distribution $\mathcal N(0, \Sigma)$, whether with $\Sigma = \Sigma_n$ or $\Sigma = \Sigma_{\infty}$, and similar choice for the bootstrap counterpart $\Sigma^b$. In our paper, we show that using $\Sigma = \Sigma_n$ leads to an approximation rate up to $n^{-1/2}$ as $\gamma \to 1$. The reviewer is correct that $\Sigma_n$ is close to $\Sigma_{\infty}$ as $n \to \infty$, but the error term coming from $\| \Sigma_n - \Sigma_{\infty}\|$ yields slower rates of convergence, compared to those obtained in our submission. Our lower bound (see Proposition 1) shows that choosing $\Sigma = \Sigma_\infty$ for Gaussian approximation in the "real" world leads to a worse rate for $\gamma \in (3/4,1)$.
>
> ***References***
>
> [31] Boris T Polyak and Anatoli B Juditsky. Acceleration of stochastic approximation by averaging. SIAM journal on control and optimization, 30(4):838–855, 1992.
>
> [42] Qi-Man Shao and Zhuo-Song Zhang. Berry–Esseen bounds for multivariate nonlinear statistics with applications to M-estimators and stochastic gradient descent algorithms. Bernoulli, 28(3):1548–1576, 2022.
>
> [47] Yanjie Zhong, Todd Kuffner, and Soumendra Lahiri. Online Bootstrap Inference with Nonconvex Stochastic Gradient Descent Estimator. arXiv preprint arXiv:2306.02205, 2023.

---

> > ### Comment · Reviewer_mjJN · 2025-08-04
> >
> > Thank you for the responses. These have clarified several misunderstandings on my side. I tend to agree with the other reviewers as well and this is sound work. I can raise my score to a 4.

---

### Decision · Program_Chairs · 2025-09-17

**Decision:**

Reject

**Comment:**

This paper proves the non-asymptotic validity of the multiplier bootstrap for constructing confidence sets in Stochastic Gradient Descent (SGD), avoiding covariance approximations of Polyak-Ruppert iterates. It provides a fully non-asymptotic bounds on bootstrap accuracy in SGD, achieving potentially faster rates than classical central limit results.

The reviewers felt the the assumptions of this paper to be stronger and requires careful justification. Furthermore, given the many recent (closely related works), it is important for this draft to clearly state the novelty in the proof techniques. The authors are encouraged to undertake a thorough revision taking the reviewers comments into account for their next submission.